# Differential Metabolites and Metabolic Pathways Involved in Aerobic Exercise Improvement of Chronic Fatigue Symptoms in Adolescents Based on Gas Chromatography–Mass Spectrometry

**DOI:** 10.3390/ijerph19042377

**Published:** 2022-02-18

**Authors:** Shanguang Zhao, Aiping Chi, Bingjun Wan, Jian Liang

**Affiliations:** 1Institute of Physical Education, Shaanxi Normal University, Xi’an 710119, China; zhaoshan@snnu.edu.cn; 2First Middle School of Shenmu City, Shenmu 719300, China; bokun@snnu.edu.cn

**Keywords:** chronic fatigue syndrome, adolescent, middle school student, aerobic exercise, metabolomics

## Abstract

Studies have found that the prevalence of chronic fatigue syndrome (CFS) in adolescents has continued to increase over the years, affecting learning and physical health. High school is a critical stage for adolescents to grow and mature. There are inadequate detection and rehabilitation methods for CFS due to an insufficient understanding of the physiological mechanisms of CFS. The purpose of this study was to evaluate the effect and metabolic mechanisms of an aerobic running intervention program for high school students with CFS. Forty-six male high school students with CFS were randomly assigned to the exercise intervention group (EI) and control group (CFS). Twenty-four age- and sex-matched healthy male students were recruited as healthy controls (HCs). The EI group received the aerobic intervention for 12 weeks, three times a week, in 45-min sessions; the CFS group maintained their daily routines as normal. The outcome measures included fatigue symptoms and oxidation levels. Keratin was extracted from the nails of all participants, and the oxidation level was assessed by measuring the content of 3-Nitrotyrosine (3-NT) in the keratin by ultraviolet spectrophotometry. All participants’ morning urine was collected to analyze urinary differential metabolites by the GC-MS technique before and after the intervention, and MetaboAnalyst 5.0 was used for pathway analysis. Compared with before the intervention, the fatigue score and 3-NT level in the EI group were significantly decreased after the intervention. The CFS group was screened for 20 differential metabolites involving the disruption of six metabolic pathways, including arginine biosynthesis, glycerolipid metabolism, pentose phosphate pathway, purine metabolism, β-alanine metabolism, and arginine and proline metabolism. After the intervention, 21 differential metabolites were screened, involved in alterations in three metabolic pathways: beta-alanine metabolism, pentose phosphate metabolism, and arginine and proline metabolism. Aerobic exercise was found to lessen fatigue symptoms and oxidative levels in students with CFS, which may be related to the regulation of putrescine (arginine and proline metabolism), 6-Phospho-D-Gluconate (starch and sucrose metabolism pathway), and Pentose (phosphate metabolism pathway).

## 1. Introduction

Chronic fatigue syndrome (CFS) is a syndrome in which individuals feel prolonged fatigue, combined with symptoms such as a low fever, headache, sore throat, muscle and joint pain, poor concentration, memory loss, sleep disturbance, and depression. It is estimated that the global prevalence of CFS is approximately 1 to 2% [1,2]. A population survey among adolescents indicated an annual incidence of 0.5% and a prevalence of 0.19% to 1.29% [3], with a male-to-female ratio varying from 2:1 to 3:1 [4,5]. The rate of CFS among adolescents in the US [6] and the United Kingdom [7] is between 0.19% and 2%, while that in Japan is relatively high, reaching 2.8% [8]. Adolescents with CFS often have extensive disease progression, leading to school absenteeism and long-term consequences concerning their educational and social development [9,10]. Therefore, detecting and preventing CFS in adolescents should be a top priority for a country’s future development.

Middle school (junior middle school and high middle school) is a critical period of growth and maturity for adolescents [11]. At the same time, they face the pressures of further education, long-term stressful study, insufficient exercise, and lack of sleep, making them extremely vulnerable to CFS [12]. The Chinese college admissions process involves a critical examination in China’s education system, and it is a necessary pathway for students who wish to attend college [13]. One study showed that 83% of high school students receive 5–6 h of sleep per night, and only 17% slept 7–8 h per night [14]. Shi et al. (2018) showed that the proportion of Chinese middle school students with CFS was close to 1%, and the proportion with potential CFS was more than 12% [15]. As a result of CFS, adolescents show memory loss and academic performance decline, affecting their physical and mental health development. In addition, some of the high-intensity exercise test items specified in the National Student Fitness Standards, such as the 1000-meter run (male) and the 800-meter run (female), are potentially risky for some students with CFS. During physical health tests in recent years, it has been found that in running sports activities, some students complain of discomfort and show abnormal physical reactions during or after exercise, and have even occasionally died suddenly [16]. Animal experiments have demonstrated that chronic fatigue might lead to sudden death after heavy exercise [17]. Therefore, it is crucial to detect and assess CFS in middle school students.

It has been shown that adolescents can eventually recover from CFS if they receive appropriate treatment soon after onset [18]. The main treatment methods include cognitive therapy, graded exercise therapy, and traditional Chinese medicine therapy. However, studies have reported that the effective rate of cognitive therapy is low [19], while traditional Chinese medicine therapy is complicated and takes a long time, making it unsuitable for middle school students [20]. A review suggested that exercise therapy is a promising treatment for CFS in adults [21], but studies in adolescents have not yet been reported. The conditions of exercise intervention are extremely important, mainly in terms of exercise intensity, exercise duration, and exercise frequency. Some studies have shown the positive effect of long-term appropriately graded aerobic exercise in CFS treatment [22,23]. Aerobic exercise to improve CFS regulates the body’s nervous system, improves oxidative capacity, and enhances immune function [24,25]. Multiple studies have shown that aerobic exercise, including cycling, walking, swimming, and jogging, can significantly improve fatigue symptoms in CFS, with intensity levels ranging from HR at 40% of VO_2_max to HR at 75% of VO_2_max [22,26,27]. The control of exercise intensity is carried out by measuring the heart rate. The American College of Sports Medicine has developed corresponding aerobic exercise intensity grading standards for healthy individuals: 35–54% of the maximum heart rate is low intensity, 55–69% of the maximum heart rate is medium intensity, and 70–89% of the maximum heart rate is high intensity (maximum heart rate = 220). Long-term exercise intervention not only improves chronic diseases but also enhances the physical health of secondary school students. Exercise frequency refers to the number of interventions performed per week and events per session. Research has shown that interventions in the frequency domain for CFS are generally performed 3–5 times per week for 30–50 min each time [28,29]. The effect of the intervention has been measured primarily by rating scales of symptom levels, including fatigue as measured by the Fatigue Scale and the Fatigue Severity Scale [30].

A variety of studies have been conducted to date, including studies of the immune system, the endocrine system, the gut microbiota, and the nervous system, to investigate the pathogenesis of CFS [31]. Nevertheless, as CFS symptoms are complex and diverse, involving multisystem and multiorgan lesions [32], there is no consistent explanation for the etiology of CFS and, thus, no relevant specific diagnostic criteria. The current diagnosis of CFS is usually based on questionnaire criteria revised by the U.S. Centers for Disease Control in 1994 (CDC-1994) [33], with additional medical tests for symptoms and signs and laboratory tests for physiological and biochemical indicators. Laboratory tests usually include viral infection tests, immune system tests, and neuroendocrine factors. For example, high levels of immunosuppressive cytokines, especially TGF-β [34], altered composition of the gut microbiome [35], and nanoelectronic detection can serve as potential diagnostic biomarkers for CFS [36]. However, these testing methods are relatively complex and not easy to generalize. Fingernails are often used as diagnostic samples for many diseases [37,38]. 3-Nitrotyrosine (3-NT) in nail keratin is a product of the oxidation of body tyrosine, which indirectly reflects the degree of oxidation of body proteins [39]. Studies have demonstrated that there is a high level of oxidative stress in CFS patients, which inevitably causes changes in the composition of macromolecules, such as proteins in the body, and this change further causes changes in the composition of keratin in the nails [40,41,42]. Therefore, the nails might be appropriate for detecting biometric CFS information in middle school students.

Metabolomics is a discipline that quantitatively studies the trends of metabolites in organisms in response to internal and external factors, and can reflect changes in biochemical and functional processes in an organism according to the trends of metabolites [43]. Many functions, such as energy metabolism and oxidative stress, are altered during fatigue, which is reflected in metabolites [44,45]. For example, chronic fatigue [46], exercise fatigue [47], and overexertion [48] can all lead to changes in the metabolism of small molecule compounds in the body. Metabolomics searches for abnormal metabolites that change during disease development [49]. These metabolic markers can evaluate the effectiveness of treatment, helping establish a convenient and specific evaluation tool and index system. Metabolomics studies can screen for specific markers and explore their pathogenesis in the CFS population. In terms of the selection of metabolites, urine is often used to diagnose many diseases due to the convenience of urine sampling and the lack of damage to the organism [50,51]. The technology platforms commonly used today include magnetic resonance technology and mass spectrometry [52]. However, magnetic resonance technology uses ultralow temperature probe technology, and its detection sensitivity is at the nanogram level, which is lower than that of mass spectrometry. Additionally, the dynamic detection range is limited, so it is difficult to detect components with large differences in concentration in the sample by taking measurements simultaneously [53]. Gas chromatography-mass spectrometry (GC–MS) is a widely used technique, the main advantage of which is the ability to detect samples with high resolution and sensitivity and to compare them with the library of standard spectra after detection, thus facilitating the metabolomics analysis of the samples to be measured and the qualitative detection of the experimental results through analytical screening [54].

The purpose of this study was to investigate the ameliorating effects and metabolic mechanisms of aerobic exercise in middle school students with CFS. In this study, GC–MS technology was used to identify the differential metabolites and metabolic pathways closely related to CFS by comparing the urinary metabolites of middle school students with and without CFS. Then, the mechanism underlying the effect of exercise therapy on CFS was explored from the perspective of an exercise intervention for CFS. In addition, the level of 3-NT in nail keratin was measured in all participants before and after the intervention to investigate whether this could be used as an indicator of detection and improvement in students with CFS.

## 2. Materials and Methods

### 2.1. Participants

The participants were from a high school in Shaanxi Province, China, a full-time closed management school where students with CFS are concentrated. Forty-six male high school students with CFS (mean age = 17.72 years, SD = 0.41 years) were recruited for this study according to diagnoses following the CDC-1994 diagnostic criteria by a psychiatrist [55]. According to the random number table method, the CFS students were divided into the exercise intervention group (EI) and control group (CFS), with 23 students in each group. Moreover, 24 age- and sex-matched healthy high school male students (mean age = 17.42 years, SD = 0.74 years) were recruited into healthy controls (HCs).

The CDC-1994 diagnostic criteria are as follows: (1) The main criterion is weak fatigue lasting for more than three months, which cannot be relieved by rest, excluding other explainable diseases causing fatigue. (2) Secondary criteria: at least four or more of the following: fever; sore throat; axillary tenderness; muscle pain; arthralgia; headache; sleep disturbance; tiredness lasting 24 h after activity; psychoneurological symptoms (e.g., irritability, forgetfulness, inattention, difficulty thinking, depression, etc.). (3) Objective criteria: at least two of the following signs: hypothermia (oral temperature 37.6~38 °C); nonexudative pharyngitis; pain in the throat for a long period; mild enlargement of lymph nodes in the neck or axilla with tenderness. The inclusion criteria for CFS were as follows: (1) met the above diagnostic criteria; (2) age between 15 and 18; (3) agreed to exercise and had not received medical treatment for chronic fatigue syndrome; and (4) voluntarily participated in this study and signed their informed consent. The exclusion criteria were as follows: (1) fatigue relieved by rest and insufficient to cause a substantial decrease in the ability to live and learn; and (2) fatigue results in the inability to exercise as required.

All subjects volunteered to participate in this experiment and received permission from their guardians and schools. All participants provided written informed consent before we collected urine samples. The height, weight, and BMI of the participants were obtained using a body composition analyzer (Inbody260, Seoul, Korea). The study was conducted following the Declaration of Helsinki, and research ethics approval was obtained from the Ethics Committee of Shaanxi Normal University (Approval ID: 202116011).

### 2.2. Exercise Intervention Program

The design of the aerobic exercise program was based on the physical condition of the students with CFS and the guidelines of the American College of Sports Medicine. The exercise intervention group received a 12-week aerobic training program with three supervised workouts per week. The total duration of exercise was no less than 45 min, with each session consisting of a 5-min warm-up, 20–30 min of aerobic running, and a 10-min relaxation period. Heart rate was monitored by wearing an activity band.

(1)Warm-up phase: Active movement of the six major joints and jogging for 5 min with a heart rate no higher than 55% HRmax (HRmax = [(220-age) × 55%]).(2)Aerobic running phase: Under the guidance of the physical education teacher, the participants gradually increased their speed so that they reached the prescribed exercise intensity of 55~65% HRmax (HRmax = [(220-age) × 55~75%]) within 10 min, and all subjects maintained this exercise intensity for 20 min. If the subjects could not complete 20 min continuously, they were given rest time until they reached 20 min cumulatively (except in the case of discomfort or injury).(3)Relaxation phase: Subjects performed 10 min of slow walking or stretching activities.

### 2.3. Fatigue Level Measurement

In this study, the fatigue level of students was used as the evaluation index of the effect of the exercise intervention. The fatigue scale 14 (FS-14) developed by [56] of the British Institute of Psychological Medicine is widely used to measure the severity of fatigue. The 14-item scale assesses two dimensions of chronic fatigue: physical fatigue (Items 1–8) and mental fatigue (Items 9–14). The participant selects “yes” or “no” for each item, with “yes” being scored as 1 and “no” as 0. The total fatigue score is calculated by summing all items. The Chinese version of the Fs-14 has good reliability and validity [57]. The evaluation was carried out before and after the exercise intervention.

### 2.4. Oxidation Level Detection

#### 2.4.1. Sample Handling

Keratin was extracted from the nails of all participants, and the oxidation level was assessed by measuring the content of 3-NT in the keratin by ultraviolet spectrophotometry. We washed the samples (40 μg) with 5 mL 2% sodium dodecyl sulfate (SDS) and 50 mmol/L sodium phosphate (pH 7.8). After drying, they were re-immersed in a 5 mL 2% SDS mixture, 50 mmol/L sodium phosphate (pH 7.8), and 20 mmol/L dithierythritol (DTE), and incubated overnight at 65 °C. The samples were stirred at room temperature with magnetic stirrers for 1 h and centrifuged (15,000× *g*) for 20 min. The samples that did not reach dissolution were re-treated 4–5 times according to the above steps, and the supernatant was combined and tested.

#### 2.4.2. Protein Standard Curve and Total Protein Assay

The standard curve of protein content was plotted with different amounts of bovine serum albumin (BSA) (0–40 μg). We diluted the supernatant appropriately and determined the absorbance value (A595). The corresponding protein content was read on the resulting standard curve.

#### 2.4.3. Separation of Proteins

Protein separation was performed by sodium dodecyl sulfate-polyacrylamide gel electrophoresis (SDS-PAGE), using 12% flat gel. Staining was done with Coomassie brilliant blue R 250, 10% acetic acid, and 40% ethanol for 1 h. The color was decolorized with 10% acetic acid and 40% ethanol. Proteins separated by SDS-PAGE were transferred to a nitro fiber membrane in transfer buffer, sealed with blocking solution at room temperature, and immunoblotted with polyclonal anti-3-NT antibody. Then, a sheep anti-rabbit antibody labeled with horseradish peroxidase was incubated at room temperature for one hour, and the nitrosamine 3-NT was detected by chemiluminescence. The membrane was scanned, and the molecular mass and net optical density values of the target bands were analyzed using a gel image processing system. Western blotting results were analyzed and processed by Image J software.

### 2.5. Metabonomics Analysis

#### 2.5.1. Urine Sample Collection and Extraction of Metabolites

All participants were asked to abstain from smoking, alcohol, and drugs for one week before sample collection. It has been shown that consuming a standard diet before sample collection reduces variation in urinary metabolic profiles in individuals [58]. Therefore, participants were provided with a standard diet for three days before urine samples were taken. Morning urine was collected one day before and one day after the intervention. After an overnight fast, the 2 mL morning midstream urine samples (7:00–8:00 a.m.) were collected with a urine collection tube with a lid for all participants, and transported in liquid nitrogen tanks at −80 °C for cryopreservation.

Then, 100 µL of the samples was transferred to an EP tube, and 20 μL of urease (80 mg/mL stock in dH2O) was added and incubated for 1 h at 37 °C. After adding 350 μL of methanol, 10 μL of internal standard (L-2-chlorophenyl alanine, 1 mg/mL stock) was added. This was followed by ultrasonication for 10 min in ice water. After centrifugation at 4 °C for 15 min at 10,000 rpm, 200 μL of supernatant was transferred to a fresh tube. To prepare the quality control (QC) sample, 50 μL of each sample was removed and combined. The QC samples were inserted regularly and analyzed in every 10 samples. After evaporation in a vacuum concentrator, 80 μL of methoxyamination hydrochloride (20 mg/mL in pyridine) was added. Then the samples were incubated at 80 °C for 30 min and derivatized by 100 μL of BSTFA reagent (1% TMCS, *v*/*v*) at 70 °C for 1.5 h. Samples were gradually cooled to room temperature, and 10 μL of fatty acid methyl ester (in chloroform) were added to the QC sample. Then, all samples were analyzed by gas chromatography coupled with a time-of-flight mass spectrometer (GC-TOF-MS).

#### 2.5.2. GC-TOF-MS Analysis

GC-TOF-MS analysis was performed using an Agilent 7890 gas chromatography coupled with a time-of-flight mass spectrometer. The system utilized a DB-5MS capillary column. A 1 μL aliquot of sample was injected in splitless mode. Helium was used as the carrier gas, the front inlet purge flow was 3 mL min^−1^, and the gas flow rate through the column was 1 mL min^−1^. The initial temperature was kept at 50 °C for 1 min, raised to 310 °C at a rate of 10 °C min^−1^, and then kept for 8 min at 310 °C. The injection, transfer line, and ion source temperatures were 280, 280, and 250 °C, respectively. The energy was −70 eV in electron impact mode. The mass spectrometry data were acquired in full-scan mode with an *m*/*z* range of 50–500 at a rate of 12.5 spectra per second, after a solvent delay of 6.33 min.

#### 2.5.3. Metabolomics Data Pre-Processing

The data analysis, including peak extraction, baseline adjustment, deconvolution, alignment, and integration, was finished with Chroma TOF (V 4.3x, LECO) software. It mainly included the following steps: (1) Individual peaks were filtered to remove noise. Filtering of deviations was performed based on the relative standard deviation. (2) Individual peaks were further filtered. Only peak area data with no more than 50% null values in a single group or no more than 50% null values in all groups were retained. (3) The missing values in the original data were simulated. The numerical simulation method was the minimum value one-half method for filling. (4) Data normalization was performed using the total ion current (TIC) of each sample. Then, the LECO-Fiehn Rtx5 database was used for metabolite identification by matching the mass spectrum and retention index. Finally, the peaks detected in less than half of the QC samples or RSD > 30% in the QC samples were removed [59].

#### 2.5.4. Multivariate Pattern Recognition Analysis

The processed data were fed into SIMCA + 14.1 software (V14.1, Umetrics AB, Umea, Sweden) for multivariate statistical analysis, including principal component analysis (PCA) (Wiklund et al., 2008) and orthogonal projections latent structures-discriminate analysis (OPLS-DA) (Worley & Powers, 2015). First, the QC and experimental samples were analyzed using PCA to ensure uniform distribution among samples and stability throughout the analysis process. PCA is a statistical method that converts a set of observed, potentially correlated variables into linearly uncorrelated variables by orthogonal transformation. GLC-based metabolomics data have the characteristics of high dimensionality (many metabolite species detected) and small sample size (small sample size detected), which contain both differential variables associated with categorical variables and a large number of non-differential variables correlated with each other. This leads to the fact that if we use the PCA model for analysis, the different variables are scattered over more principal components due to the influence of the correlated variables, preventing better visualization and subsequent analysis. Second, we used OPLS-DA analysis to filter out the orthogonal variables in metabolites that were not correlated with categorical variables and analyzed the non-orthogonal and orthogonal variables separately to obtain more reliable information about the group differences in the metabolites. To evaluate the validity of the OPLS-DA model, the quality of the model was tested by 7-fold cross-validation. The *R*^2^*Y* (interpretability of the model) and *Q*^2^ (predictability of models) were used to judge the model validity. The model validity was further tested by a permutation test, in which the order of the categorical variable *Y* was randomly changed several times (*n* = 200) to obtain different random *Q*^2^ values.

### 2.6. Statistical Analysis

The variable importance in projection (VIP) value of each variable in the PLS-DA model was calculated to indicate its contribution to the classification. Metabolites with VIP > 1 were further applied to Student’s *t*-test at the univariate level to measure the significance of each metabolite, with results adjusted for multiple testing using the Benjamini–Hochberg procedure, with the critical false discovery rate (FDR) set to 0.05. The metabolic pathways were analyzed by MetaboAnalyst 5.0 (Shanghai Luming Biotechnology Co., LTD, Shanghai, China).

All statistical analyses were performed by SPSS (23.0; SPSS, Inc., Chicago, IL, USA). The Shapiro–Wilk test was used to determine the normality of the data distribution. Continuous variables are expressed as the means with SD or medians with interquartile ranges. Differences at baseline for the demographic information, 3-NT, and reported fatigue symptoms between the two groups were compared using an independent-samples *t*-test for continuous data. The intragroup differences between before and after intervention for 3-NT and reported fatigue symptoms were compared using a pairwise *t*-test. A 2 (group: EI, CFS) × 2 (times: baseline, 12 weeks) × 3 (total figure, physical fatigue, mental fatigue) mixed repeat measure of covariance (ANCOVA) was performed to test the interaction effect of time and group for FS-14 outcome. A Greenhouse–Geisser correction of the ANOVA assumption of sphericity was applied where appropriate. The Bonferroni-correction method was used to correct multiple comparisons. The significance level was set as a 2-sided *p* value of less than 0.05.

## 3. Results

### 3.1. Comparison of General Information

Twenty participants (87.0%) in the EI group and twenty-three (100%) in the CFS group completed the program. Three participants in the EI group withdrew from the study. There was no statistically significant difference in the general data for age, BMI, and education level among the participants in each group (*p* > 0.05) (Table 1).

### 3.2. The Efficacy of Exercise Intervention

The main effect of groups was significant, *F* (1, 42) = 213.62, *p* < 0.001, *η*^2^ = 0.84. The main effect of intervention times was significant, *F* (1, 42) = 813.31, *p* < 0.001, *η*^2^ = 0.95. The main effect of the fatigue symptoms was significant, *F* (2, 84) = 1048.14, *p* < 0.001, *η*^2^ = 0.96. The interaction effect of intervention times and groups, intervention times and fatigue symptoms, and groups and fatigue symptoms were also significant, *F* = 175.09, 23.02, 49.76, *p*’*s* < 0.001, *η*^2^ = 0.81, 0.35, 0.54. Simple effect analysis was performed on the fatigue symptoms. The results of post hoc tests indicated total fatigue score (*p* < 0.001), physical fatigue score (*p* < 0.001), and mental fatigue score (*p* < 0.001) in the EI group were significantly improved at 12 weeks than baseline values. Simple effect analysis was performed on the intervention time. The results of post hoc tests revealed that total fatigue score, physical fatigue score, and mental fatigue score were significantly improved in the EI group compared to the CFS group at 12 weeks (*p*’*s* < 0.001). Intra- and intergroup differences in subjective outcome measures, including total fatigue scores, physical fatigue scores, and mental fatigue scores, are shown in Table 2.

The 3-NT content in the nails was significantly higher for the CFS group than for the HCs group before the intervention (*p* < 0.05), suggesting an increased level of oxygenation stress in the CFS students. Compared with the CFS group, there was no significant difference in the EI group before the exercise intervention, while after the exercise intervention, the 3-NT content of the EI group significantly decreased (*p* < 0.05) (Figure 1).

### 3.3. Multivariate Data Analysis Results

The total ion chromatogram of each group is shown in supplementary materials Appendix A, where the abscissa represents retention time, and the ordinate represents ion flow intensity. The 67,432 peaks were extracted in the metabolic profiling. After a data processing series, 44,556 peaks were retained for MetaboAnalyst analysis (www.metaboanalyst.ca, accessed on 24 June 2020). Then, logarithmic transformation and pareto scaling were performed on these features. Finally, 1411 substances were qualified.

We first established the PCA and OPLS-DA models with a supervised pattern recognition method for the HCs and CFS groups. Figure 2 shows the graph of the metabolomics dataset’s metabolic profiles from the PCA. It can be seen from the PCA score plot for all samples that the sample was basically within the 95% confidence interval (Hotelling’s T-Squared Ellipse). The relatively clustered QC samples showed that the system was reproducible and that the method was stable and reliable. Further analysis was required, as the degree of separation was not clear. The OPLS-DA model indicated that the CFS group was significantly separated from HCs, with less overlap, cumulative *R*^2^*X* at 0.26, *R*^2^*Y* at 0.97, and *Q*^2^*Y* at 0.77 (Figure 3A). Then, the OPLS-DA model for the EI and CFS groups was constructed after the intervention. The model indicated that the EI group was significantly separated from the CFS group with little overlap, with cumulative *R*^2^*X* at 0.21, *R*^2^*Y* at 0.92, and *Q*^2^*Y* at 0.64 (Figure 3B). In addition, the permutation test showed that the two OPLS-DA models were robust, without overfitting (Figure 3).

### 3.4. Differential Metabolite Results

All compounds were screened for differential metabolites by OPLS-DA analysis through the KEGG database. The differential metabolites (CFS vs. HCs and EI vs. CFS) were screened using a combination of multivariate and univariate statistical criteria (VIP > 1 and FDR < 0.05). Before the intervention, there were 20 differential metabolites between the CFS and HCs groups. The results show that, compared with the HCs group, the changes in the relative levels in the CFS group were as follows: except for 4-hydroxyphenyl acetic acid, inosine, 6-phosphoglucose acid, N-carbamylglutamate, and L-dithiothreitol, which increased significantly while the other 14 substances decreased significantly (Table 3). After the intervention, there were 21 differential metabolites between the EI and CFS groups, of which 11 showed a significant increase in relative levels (Table 4).

### 3.5. Metabolic Pathway Analysis

The differential metabolites between the HCs and CFS group and between the EI and CFS groups were input into the Metabo-Analyst database to construct metabolic pathways. The pathway impact value (*p* ≥ 0.1) of the pathway enrichment topology analysis was taken as the main target pathway. Compared with the HCs group, five metabolic pathways were disturbed in the CFS group, including beta-alanine metabolism, arginine biosynthesis, glycerol metabolism, pentose phosphate metabolism, and arginine and proline metabolism (Figure 4a). The aerobic exercise intervention affected four metabolic pathways in CFS students, compared to before the exercise intervention: arginine and proline metabolism, pentose phosphate metabolism, and starch and sucrose metabolism (Figure 4b).

## 4. Discussion

It is well known that exercise and physical activity can improve the health and quality of life of individuals with chronic diseases [60,61,62]. Although there is no definitive treatment for CFS, many researchers believe that appropriate amounts of exercise are needed for those with CFS [29,63,64]. The results of this study suggested that 12 weeks of aerobic running improved fatigue symptoms in male students with chronic fatigue, as evidenced by a significant change in the total fatigue score. A study used 30 min of exercise per day for 12 weeks at 60% maximal oxygen uptake to treat CFS and showed that 55% of patients felt significantly better after the self-perceived health status assessment [65]. Our results further suggest that aerobic exercise has an ameliorating effect on both physical fatigue symptoms and mental fatigue symptoms. The related release of β-endorphin and dopamine induced by exercise provides relaxation and pleasure effects in regular practitioners [66].

To investigate the metabolic mechanism of aerobic exercise to improve chronic fatigue, urine samples from male students with CFS and healthy students were screened for differential metabolites using GC–MS metabolomics techniques. A total of 21 differential metabolites were detected (Table 2), and increases or decreases in these metabolites may be closely related to the pathogenesis of CFS in high school students. To better elucidate the mechanism of CFS pathogenesis, we constructed a metabolic pathway map of differential metabolites, which involved five metabolic pathways, including alanine metabolism, arginine biosynthesis, glycerophospholipid metabolism, and pentose phosphate metabolism (Figure 4a). Aerobic exercise may improve fatigue symptoms by interfering with the above metabolic pathways. These pathways are mainly related to amino acid metabolism, lipid metabolism, and glucose metabolism. Therefore, the regulation of amino acid metabolism, lipid metabolism, and glucose metabolism disorders are possible mechanisms by which exercise improves fatigue symptoms (Figure 4b).

CFS is a more severe form of persistent fatigue, with symptoms such as fatigue, pain in certain parts of the body, memory and cognitive decline, and poor resistance to illness [67]. Studies have reported that CFS symptoms are associated with energy metabolism disorders [68], immune dysfunction [69], and hypothalamic–pituitary–adrenal (HPA) axis hypofunction [70]. In the previous phase, our team studied the peripheral differential metabolites and related metabolic pathways in CFS rats using non-targeted metabolomics. We preliminarily identified metabolic pathways associated with CFS, including energy metabolism, amino acid metabolism, pyrimidine metabolism, sphingolipid metabolism, and steroid and antibiotic synthesis. Among the metabolic pathways involved in CFS, disorders include the tricarboxylic acid metabolic pathway related to energy metabolism, glutamate-alanine-aspartate metabolic pathway related to immunity, sphingolipid metabolic pathway related to cognition, and steroid metabolic pathway related to HPA axis dysfunction [47]. Amino acids are the basic units of proteins essential to the organism’s immune system [71]. We found that reduced glutamate levels and higher citrulline levels in the urine of male students with CFS resulted in disorders of arginine and proline metabolism. Glutamate is an essential transaminase amino acid, and its catabolic compounds are used for aerobic respiration in the tricarboxylic acid cycle. The decrease in acetyl glutamate levels is a sign of a lower level of aerobic respiration of the body, leading to a decrease in oxygen utilization in the mitochondria and ultimately an increase in the concentration of oxygen in the blood [72]. Arginine is converted to citrulline, catalyzed by nitric oxide synthase in the case of an increased blood oxygen concentration that produces nitric oxide [73]. Therefore, the increased level of citrulline in the urine of CFS students invariably produces large amounts of nitric oxide. Nitric oxide combines with superoxide to form reactive nitrogen species (RNS) [74].

The excessive production of RNS leads to the disruption of the balance of the antioxidant system and increases oxidative stress in the body. High RNS levels lead to DNA damage, and a decrease in choline levels in serum leads to DNA strand breakage, which induces apoptosis [75]. Free radicals can reduce the fluidity of the cell membrane and affect cellular Ca2+ transport, which ultimately leads to cellular damage and decreased cellular function, resulting in fatigue of the body [76]. We found that the level of 3-NT in the nails was significantly higher for the CFS group than the HCs group, indicating that the oxidative system in students with CFS was stronger than the antioxidant protection system (Figure 1). Studies in animal models have shown that when organisms are under severe oxidative stress, RNS increases the level of 3-NT in the body by causing tyrosine nitrosylation, which then directly causes protein and enzyme denaturation, DNA damage, and finally apoptosis or cell death [77]. The mitochondria in the human body contain important proteins or enzymes such as ATPase, acetyl coenzyme A, and cytochromes, but the tyrosine in these substances is nitroxylated, and respiratory chain disorders, tricarboxylic acid cycle abnormalities, and energy metabolism problems may occur. A study found that oxidative stress indicators were higher in CFS patients than in healthy volunteers, while biological antioxidant potential indicators were lower [78]. Oxidative stress and energy metabolism have been elucidated as dysfunctions in the metabolic pathways of CFS patients [79,80].

Improving the body’s antioxidant capacity and scavenging free radicals has a huge effect on alleviating fatigue. In recent years, many studies have shown that regular aerobic exercise training contributes to eliminating oxidative stress [81,82,83]. On the one hand, aerobic exercise can reduce the content of free radicals and lipid peroxidation products in the blood, muscle, liver, and other tissues and organs after exercise, and reduce the degree of damage caused by free radicals by blocking the chain reaction of lipid peroxidation, reducing lipid peroxidation damage, and inhibiting lipid peroxidation production. On the other hand, aerobic exercise decreases the basal production of free radicals when the body is at rest, and reduces the peak value of free radicals during exercise [84]. After aerobic exercise intervention, the level of 3-NT in the EI group was significantly lower than that in the CFS group (Figure 1).

The significant downregulation of citrulline levels and upregulation of proline and glutamate levels in students with CFS under the effect of aerobic exercise intervention indicated that the hydrolysis of arginine to urea and ornithine was restored. Spermine in the spermidine and proline metabolic pathway is produced from putrescine and S-adenosylmethionine catalyzed by various enzymes, providing greater stability and flexibility to DNA molecules, with a cell proliferation-promoting cell function [85]. Therefore, the ameliorative effect of aerobic exercise on fatigue in students with CFS may be related to the enhanced ability of the body to scavenge free radicals. Arginine and proline metabolism are involved in regulating oxidative stress in the organism [86], which may be one of the metabolic pathways for exercise to improve fatigue symptoms (Figure 5).

We found that the starch and sucrose metabolism pathway was disordered in CFS students (Figure 6a). Glucose is the main energy source for cells. There are three main pathways to provide energy for the body, including aerobic oxidation, anaerobic glycolysis, and gluconeogenesis [87]. Pyruvate is a common pathway of aerobic and anaerobic metabolism [88]. When the body is sufficiently oxygenated, pyruvate enters the cell mitochondria and participates in the trisaccharide cycle, where it is completely oxidized to CO_2_ and water, while in the absence of oxygen, pyruvate is converted to lactate. In this study, no increased lactate in the urine of CFS students was detected, but rather an increase in alanine was detected (Table 2), resulting in a disruption of the alanine metabolic pathway. The main roles of alanine in the body include synthesizing tissue proteins and deaminating pyruvate and glutamate. Alanine aminotransferase, which is related to alanine, is vital to human metabolism, speeding up the conversion of proteins into amino acids in the body [89]. When the body’s tissues and organs are overactive or have certain pathologies, alanine aminotransferase will be released into the blood in large quantities, and the level of alanine aminotransferase increases, thus increasing the rate of conversion of amino acids to pyruvate and alanine [89]. A recent study finds that alanine transaminase levels are significantly higher in patients with CFS, which may be because the body needs to take in large amounts of alanine from the blood during the development of CFS to promote glucose metabolism for energy supply, enhancing autoimmunity [90]. Several studies have shown that the onset of CFS is associated with decreased immune function [91,92].

Alanine also participates in the glucose-alanine cycle for gluconeogenesis, which regulates blood sugar and makes ammonia nontoxic to transport, while providing energy for muscle activity [93]. The increased levels of alanine in the urine of CFS students indicate that the glucose-alanine metabolic pathway is abnormally active. One of the core symptoms of CFS is unexplained muscle weakness, possibly due to an increase in energy consumption that makes sufferers feel tired [94]. Aerobic exercise enables the activation of immune factors and increases the expression of growth factors and the number of immune cells, resulting in stronger immune function and reduced flu-like symptoms, such as fever and sore throat [95]. The results reflected in this experiment are consistent with the concept of CFS, in which patients often feel unexplained persistent fatigue. The symptoms are consistent with the finding that CFS causes disorders of energy metabolism in animal experiments.

This study shows that the pentose phosphate metabolic pathway may be altered in students with CFS (Figure 6b). Glucose performs anaerobic enzymolysis in the pentose phosphate pathway [96]. Glucose is decomposed into dihydroxyacetone phosphate and GAP through a series of enzymatic reactions and is converted into GAP in the form of a glycolysis pathway or aerobic oxidation under the catalysis of propanone phosphate isomerase. The pentose phosphate pathway is another pathway in glucose metabolism that starts with glucose 6-phosphate and is converted in the cytoplasm by a series of enzymes to dihydroxyacetone phosphate and phosphate ribose [97]. Under specific conditions, the two pathways can be linked and interconverted by glyceraldehyde-3-phosphate and fructose-6-phosphate, which serve as the key nodes of the glycolytic pathway and pentose phosphate pathway. The present study speculates that the reduced urinary levels of d-gluconate 6-phosphate and d-glycerol in CFS patients probably result from disturbances in the gluconeogenic metabolic pathway. The content of 6-phospho-D-gluconate in the urine of CFS students was significantly increased after aerobic exercise intervention.

There are several limitations in the current study. Although differential metabolites were obtained in this study, the metabolic pathway results showed marginal significance. In future studies, a combination of targeted and untargeted metabolomics techniques will be used to explore the metabolic mechanisms of CFS. Next, only male students were selected as the study participants. The menstrual period in females can affect the consistency of exercise interventions, and it can also affect the uniformity of metabolic outcomes. Therefore, in future studies, it will be important to consider the specificity of the female menstrual cycle and develop an exercise intervention program that is consistent with individual needs. Thirdly, the limitation of this study is the small sample size. In the future, we should expand the sample size and generalize the exercise program to reduce CFS incidence in adolescents.

## 5. Conclusions

CFS has become one of the major problems affecting human health in the 21st century. In this study, we found that 12 weeks of aerobic running could alleviate the symptoms of chronic fatigue and decline the oxidative level of the body in male high students. The metabolomic results indicated that the improvement of chronic fatigue symptoms might be related to the regulation of putrescine (arginine and proline metabolism) and 6-Phospho-D-Gluconate (starch and sucrose metabolic pathway and pentose phosphate metabolic pathway). It is important to schedule daily exercise for high school students.

## Figures and Tables

**Figure 1 ijerph-19-02377-f001:**
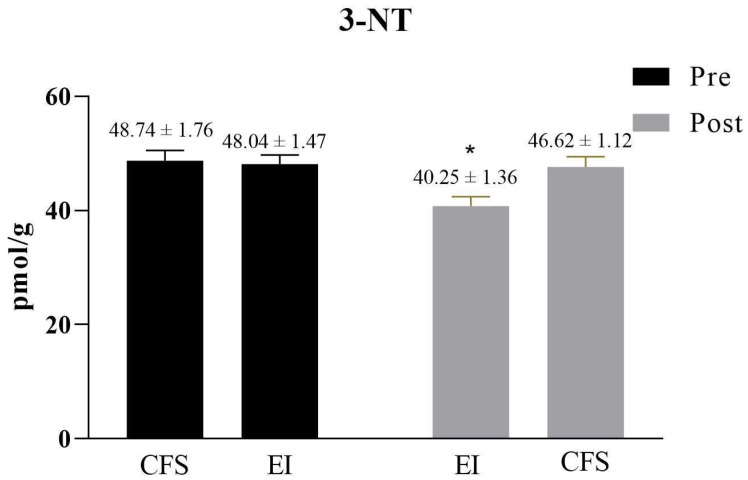
3-NT content of nails before and after the intervention. Note: * indicates statistical significance (*p* < 0.05).

**Figure 2 ijerph-19-02377-f002:**
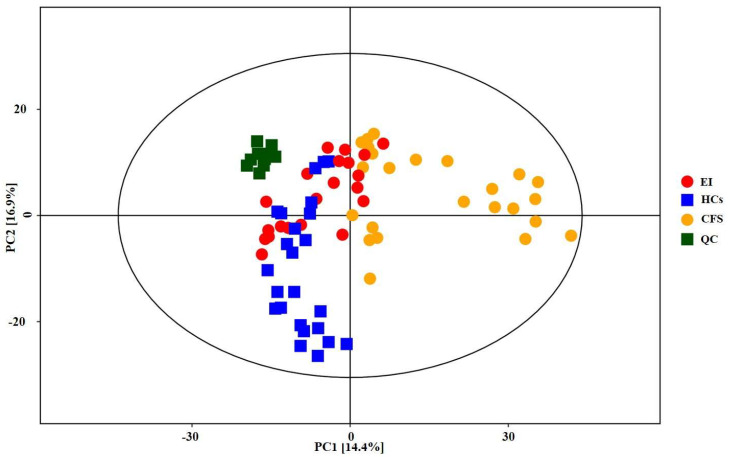
PCA scores plot of urine samples from all samples.

**Figure 3 ijerph-19-02377-f003:**
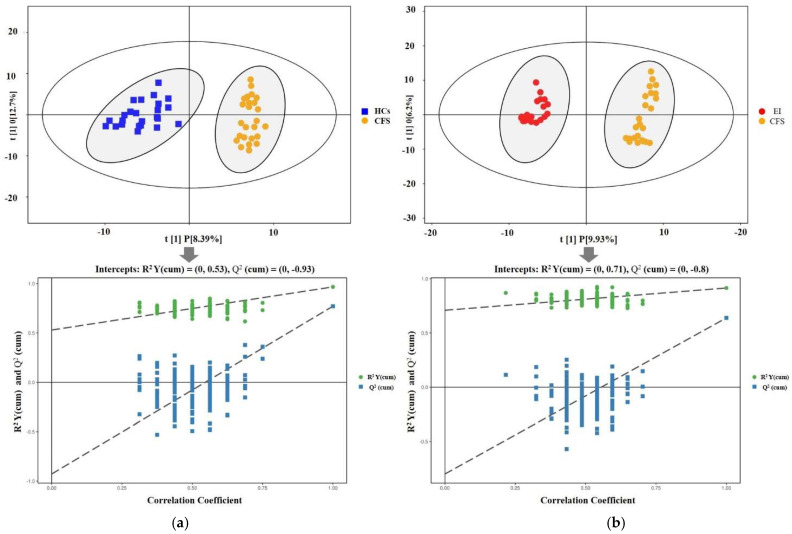
OPLS-DA scores plot and validation of OPLS-DA model by permutation test. (**a**) CFS vs. HC (**b**) EI vs. CFS. The above scatter plot of the OPLS-DA model shows obvious discrimination between CFS students (blue box) versus HCs (yellow ball) in the baseline period and CFS versus EI (red ball) after the intervention. The following is the permutation test chart. Validation of OPLS-DA model of urinary samples by permutation test (the *x*-axis means the correlation coefficient between the original *y* variable and the permutated *y* variable), and the *y*-axis is the value of R^2^ and Q^2^.

**Figure 4 ijerph-19-02377-f004:**
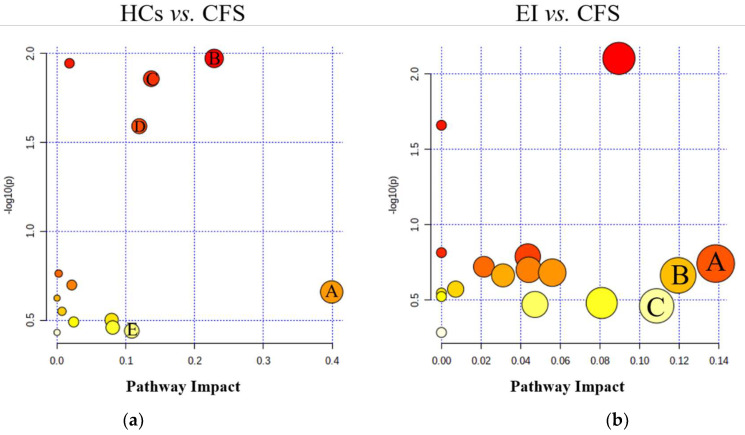
Topological score results of the metabolic pathway. (**a**) is a comparison of the HCs and CFS groups: A beta-alanine metabolism; B arginine biosynthesis; C Glycerolipid metabolism; D Pentose phosphate pathway; E Arginine and proline metabolism. (**b**) is a comparison of the EI and CFS groups after the intervention: A Starch and sucrose metabolism; B Pentose phosphate metabolism; C Arginine and proline metabolism.

**Figure 5 ijerph-19-02377-f005:**
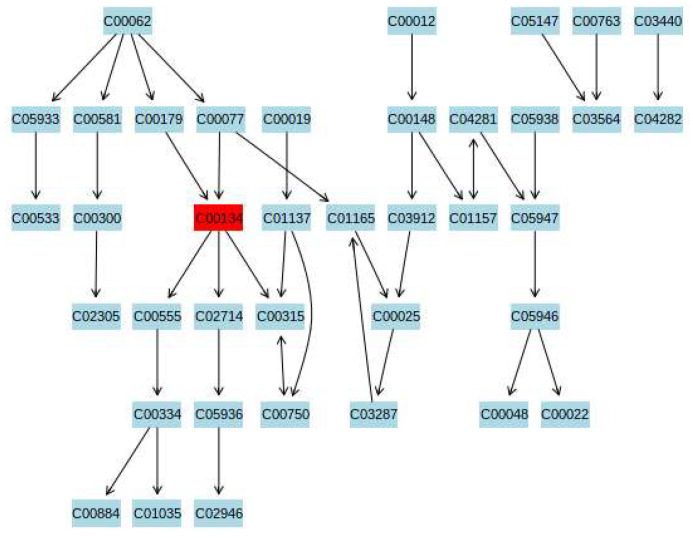
Arginine and proline metabolism pathway. C00134 Putrescine; 1,4-Butanediamine; 1,4-Diaminobutane; Tetramethylenediamine; Butane-1,4-diamine.

**Figure 6 ijerph-19-02377-f006:**
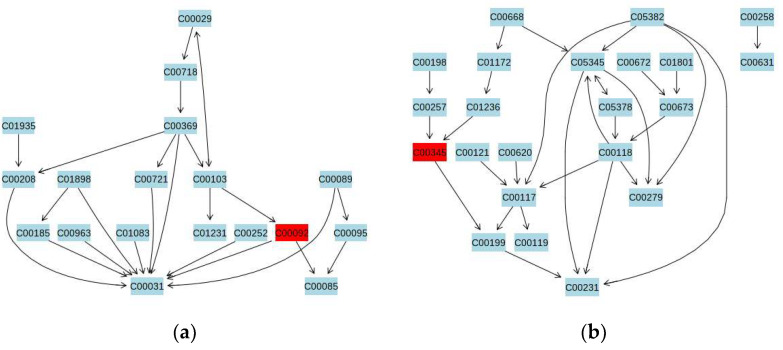
(**a**) Starch and sucrose metabolism pathway and (**b**) Pentose phosphate metabolism pathway (C00345/C00092 6-Phospho-D-gluconate).

**Table 1 ijerph-19-02377-t001:** Demographic information of participants (Mean ± SD).

Variable	CFS (*n* = 23)	EI (*n* = 23)	HCs (*n* = 24)
Age, years	16.72 ± 0.41	16.57 ± 0.28	16.42 ± 0.74
Height, cm	176.30 ± 1.37	174.50 ± 1.24	176.50 ± 1.47
Weight, kg	71.46 ± 1.73	69.24 ± 2.41	68.65 ± 3.58
BMI, kg/m^2^	22.25 ± 1.12	22.46 ± 1.31	22.92 ± 0.60

Note: CFS, chronic fatigue syndrome group; HCs, healthy controls; EI, exercise intervention group.

**Table 2 ijerph-19-02377-t002:** FS-14 results between the CFS and EI groups before and after the intervention.

	Baseline		12 Weeks	
	EI	CFS	CFS	EI
Total fatigue score	10.72 ± 0.14	10.14 ± 0.14	8.46 ±0.16	4.36 ± 0.16 ***###
Physical fatigue score	5.18 ± 0.12	5.05 ± 0.18	4.41 ± 0.22	2.13 ± 0.15 ***###
Mental fatigue score	5.22 ± 0.18	5.09 ± 0.12	4.05 ± 0.15	2.22 ± 0.22 ***###

Note: ### indicates compared with EI group at 12 weeks, *p* < 0.001; *** indicates compared with baseline, *p* < 0.001.

**Table 3 ijerph-19-02377-t003:** Differential metabolites in the CFS group compared within the HCs (group CFS vs. group HCs).

Metabolite	HMDB ID	RT (min)	*m*/*z*	VIP	*p*-Value	FC
Uric acid	HMDB0000289	20.02	55	3.17	0.00001	3.35
D-Glyceric acid	HMDB0000139	11.34	71	1.36	0.01121	2.00
Lactose	HMDB0041627	24.98	64	2.56	0.00001	2.68
4-hydroxyphenylacetic acid	HMDB0000020	15.31	179	2.3	0.00009	0.45
N-Acetyl-L-glutamic acid	HMDB0001138	15.74	117	2.87	0.00000	8.47
3,4-Dihydroxyphenylacetic acid	HMDB0001336	17.36	179	2.68	0.00003	3.29
N-Acetylglutamic acid	HMDB0001138	16.89	84	2.89	0.00000	2.91
D-(glycerol 1-phosphate)	HMDB0000126	16.60	292	2.98	0.00000	5.61
Inosine	HMDB0000195	23.99	230	1.26	0.00826	0.38
Purine riboside	HMDB0029956	22.35	103	3.02	0.00000	5.36
Beta-Alanine	HMDB0000056	12.57	174	2.06	0.00192	2.73
6-phosphogluconic acid	HMDB0001316	22.70	217	2.50	0.00014	5.47
N-Carbamylglutamate	HMDB0015673	19.46	174	1.44	0.01197	0.36
4-Androsten-11beta-ol-3,17-dione	HMDB0006773	26.82	361	1.29	0.01217	0.32
Adrenaline	HMDB0000068	17.57	267	2.84	0.00262	2.35
L-Dithiothreitol	HMDB0013593	15.00	195	2.05	0.00057	0.03
Citrulline	HMDB0000904	16.52	71	1.45	0.01879	1.52
Putrescine	HMDB0001414	16.59	72	2.87	0.00000	2.50
Elaidic acid	HMDB0000573	21.29	131	1.45	0.01773	1.52
2,2-Dimethylsuccinic Acid	HMDB0002074	11.27	71	2.70	0.00000	2.81

Note: HMDB ID, the code in the HMDB database corresponding to the substance; RT, retention time of chromatographic peak; *m*/*z*, mass-to-charge ratio; VIP, the variable projection importance of the substance obtained in the OPLS-DA model compared in this group; FC, fold change refers to the ratio of the average metabolite; FC > 1, up-regulated; FC < 1, down-regulated.

**Table 4 ijerph-19-02377-t004:** Information on differential metabolites after 12 weeks of exercise intervention (group EI vs. group CFS).

Metabolite	HMDB ID	RT (min)	*m*/*z*	VIP	*p*-Value	FC
Oxalic acid	HMDB0002329	8.58	79	1.68	0.001044	0.76
Cellobiose	HMDB0000055	24.96	79	1.00	0.014957	1.48
Uric acid	HMDB0000289	20.02	55	1.90	0.000031	3.65
Quinolinic acid	HMDB0000232	16.29	79	2.15	0.000772	0.63
N-Acetylmannosamine	HMDB0001129	19.78	79	2.27	0.000533	0.64
Lactose	HMDB0041627	24.98	38	2.38	0.000020	16.9
Orotic acid	HMDB0000226	16.44	79	1.50	0.014485	0.74
Glycerol 1-phosphate	HMDB0000126	16.60	46	2.23	0.009390	2.99
L-Malic acid	HMDB0000156	13.42	76	1.05	0.035540	1.58
Malonic acid	HMDB0000691	9.51	79	1.31	0.021348	1.36
3-Methylglutaric Acid	HMDB0000752	12.66	67	1.65	0.010768	0.62
Purine riboside	HMDB0029956	22.35	48	2.97	0.000000	19.9
Uridine	HMDB0000296	23.06	79	2.21	0.000191	0.72
6-phosphogluconic acid	HMDB0001316	22.70	32	1.97	0.002571	4.19
3-Hydroxypyridine acid	HMDB0013188	8.74	71	1.18	0.021134	1.62
Citraconic acid	HMDB0000634	12.84	76	1.05	0.002329	0.65
5,6-dihydrouracil	HMDB0000076	15.10	41	1.08	0.028759	0.35
Putrescine	HMDB0001414	21.95	57	2.11	0.000006	0.22
Glucose-6-phosphate	HMDB0001401	23.43	39	1.74	0.018920	0.22
2,2-Dimethylsuccinic Acid	HMDB0002074	11.27	31	2.18	0.001279	0.48

Note: HMDB ID, the code in THE HMDB database corresponding to the substance; RT, retention time of chromatographic peak; *m*/*z*, mass-to-charge ratio; VIP, the variable projection importance of the substance obtained in the OPLS-DA model compared in this group; FC, fold change refers to the ratio of the average metabolite; FC > 1, up-regulated; FC < 1, down-regulated.

## Data Availability

The study data can be accessed from the corresponding author A.C. by request.

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
