# Peer review of "Differential Metabolites and Metabolic Pathways Involved in Aerobic Exercise Improvement of Chronic Fatigue Symptoms in Adolescents Based on Gas Chromatography–Mass Spectrometry"

_ijerph, 2022, doi:10.3390/ijerph19042377_

Round 1
Reviewer 1 Report
Authors Zhao et al. profile metabolites and pathways in a cohort of CFS students and healthy controls to evaluate the metabolic effects of aerobic exercise. Subjects with CFS were randomly assigned to the exercise intervention (EI) group, or the control group (CFS), while healthy matched controls were also evaluated. The research question has merit and the overall scientific thinking is sound. However, the study suffers from numerous issues that fundamentally undermine the conclusions made by the authors. Additionally, errors in writing and presentation persist that call the scholarship into doubt. To improve the paper, the authors should consider the following major and minor revisions.
Major Revisions:
- Lines 52-55: I understand the college entrance exam is an important milestone for many in China, but you can delete the sentence mentioning it as a pathway out of poverty. There are no references, and it is too anecdotal a statement for scientific writing. You can delete the sentence without losing any meaning in the paragraph.
- Lines 123-131: Beginning with "For" and ending with "to the organism", the authors make many claims about metabolomics and previous applications to CFS, however none of this is cited. Please make sure every statement that conveys specialized knowledge is cited.
- Why did the authors add urease to urine samples for metabolomic profiling? Adding methanol would have been sufficient for protein precipitation and metabolite extraction (although there should not be much protein in the urine). Please justify the use of urease with a reference.
- Since the researchers measured height, weight, and BMI, they should include an anthropometrics section in the methods to outline the materials used, etc.
- How often were the QC samples injected? This should be stated in the methods.
- Lines 347-349: The authors state that the peak alignment in Figure 1 shows good repeatability and reliability of measurements. However, this is a very qualitative assessment, and not very scientific. Instead, the authors should make a graph showing QC CVs and the number of metabolites detected with <5% CV, 5-10% CV, etc. Also, they refer to Figure 1 as the total ion chromatogram in the text (which is incorrect) and as the ion flow diagram in the caption (which is correct). Please change all references to the latter.
- The authors state that 67342 peaks were extracted, while 44556 peaks were processed. How many of these were identified and how many were analyzed as mass spectral features?
- Tables 2 and 3: Instead of showing the trend as up and down arrows, analyze the groups as case/control and report fold changes. That way, you can convey directionality and magnitude of change.
- With regard to the significance testing, the authors perform t-testing between CFS and HC as well as between EI and CFS. Although these may be the comparisons of interest, the authors should instead use an ANOVA-based method to analyze all groups with each other. Since there are three groups, t-testing should not be used.
- Figure 6 and related discussion: In the pathway analysis figure between HC and CFS, pathways B, C and D are significant, but they have very little pathway impact (~0.2). Similarly, in the pathway analysis figure between EI and CFS, pathways A, B, and C are marked even though they are not significant nor do they have an appreciable impact score (~0.12). However, the authors make much discussion of these findings and suggest that pathways are the underlying mechanisms of the intervention, but that is not supported by their data. The authors should mention in their results that most of the detected pathways were non-significant and the ones that were had very low to negligible impact. Relatedly, the authors should rewrite their discussion of those affected pathways as well.
Minor revisions:
- In the abstract, the authors mention that middle school and high school students were recruited, although these are not the same population in the Western world. I understand they may be bifurcated as junior middle and high middle school in China, but this terminology will confuse Western readers. Perhaps refer to all subjects as secondary school students or just as high school students.
- In the abstract, the authors mention they used MetaboAnalyst 4.0 but, in the methods, they mention using the 5.0 version. Please correct.
- There are many instances of incorrect grammar and style. These are not too frequent to deter from understanding the paper, but they are too numerous to reasonably list. I suggest the authors enlist the help of an editing service prior to resubmission.
- Line 154: What is a repeated middle school? Are you referring to the study design or to the school from which subjects were recruited? This is confusing, please address.
- When listing the protocol steps, please do so in the past tense (see Methods).
- Please make sure you are using standard unicode characters when inserting the celsius symbol.
- Line 250: what are FAMEs? Please define at first mention.
- Please italicize all proper statistics and applicable mass spectrometry terms (i.e. p, t, R2, Q2, Y, m/z, etc.). Also, do not say "ps"; instead, say "all p" or "p's".
- None of the figure numbers are in order. Figure 3 is presented first, and then figures 1, 2-6. Also, Figure 6 is referred to as Fig. 3. Additionally, Figure 5 is referred to as Figure 1C, even though none of the figures were lettered. None of the figures are appropriately referenced nor in order. Please make sure all figures are discussed and labeled in the order mentioned, with each part correctly referenced.
- There are some issues throughout the references with citation style and format, too many to list here. The authors should carefully review their references and correct any errors.
Although the study aims to satisfy a pertinent gap in the literature relating to CFS, the manuscript is currently not fit for publication. There are major flaws in the analysis plan and results reporting/discussion. However, after significant revisions are made to the above mentioned concerns, this study may be suitable for publication at the discretion of the editor.
Reviewer 2 Report
The introduction, broad and well detailed, allows you to get into the topic well.
It is always difficult to find a close correlation between the subjects and any oxidative stress that is created. In fact, very often the body responds in various ways and having a quantifiable marker is complex. There are many suggestions from the authors for the different metabolisms that could be affected by oxidative stress in CFS patients, but without direct evidence.
However, the authors' proposal is interesting both for urinalysis and for measuring the content of 3-nitrotyrosine in keratin of nails.
It could be a valid intervention to invert the two groups (EI) and (CFS), proposing to the first not to continue with physical exercise and to the second the aerobic exercises previously used in the first group. Theoretically the group (CFS) should have an improvement as in the case of the group (EI).
In summary, I think that the work, albeit with small numbers, offers good working technology and standardization.
I do not find a weakness of the work to have evaluated only the males, because only after confirming the validity of the method, will it be possible to face the problem of females, which requires greater complexity, as much more variables have to be added.
Minimal text revisions:
Line 108 missing the space between the word “microbiome” and the round bracket. The same for line 415 where the word “disease” must be detached from the round bracket.
Line 341 is talking about Figure for the first time, why do we start with Figure number 3, instead of 1?
Figure 1, although in color, is not very clear and adds nothing to the text.
Lines 280-281 the authors write: orthogonal projections latent structures-discriminate analysis abbreviated as OPLS-DA, but in line 355, 361, 366 we find PLS-DA only. Do the two abbreviations have the same meaning?
Lines 630 and 745, the abbreviation of the magazine BMJ should have all capital letters.
Lines 646 and 681 why not shorten the magazine as in the other cases?
Reviewer 3 Report
In the manuscript submitted by Zhao and colleagues, authors reported the analysis of 3-nitrotyrosine and various urinary metabolites on forty-six male high school students affected by chronic fatigue syndrome (CFS) to better clarify the metabolic impact of the aerobic running intervention. The manuscript is well written in general, and I appreciated the detailed CFS description in both the introduction and discussion. However, results are poor, not so detailed, with table and figure captions poor of explanation. Furthermore, many references are not recent and many points of this paper need to be clarified or improved.
LINE 50: This is a general concept, well known by most readers. However, this is a scientific article, so it would be better to cite a reference, such as a statistical study or review article.
LINES 52-53: The word "examination" is redundant. Please change it with another word.
LINE 55: Authors have to include a reference.
LINE 62: Reference needs to be cited.
LINE 67: Authors have to cite the reference for this study
LINE 69: Authors are reporting the presence of many studies. If there are many studies, authors should cite more recent references.
LINE 99: If authors want to report "a variety of studies have been conducted to date", they could not just cite only one reference and of twenty years ago. Authors should cite a more recent reference at least.
LINE 113: Authors should cite some references regarding the relevance of 3-nitrotyrosine as an accepted biomarker for evaluating the oxidative stress degree in the human body.
LINE 116: Authors are reporting the presence of many studies that highlighted the presence of high oxidative stress levels in CFS patients, but they cited only one study (Maes et al., 2011). The authors should cite more studies.
LINE 161: Why are healthy controls not healthy high school male students? Authors should better justify the choice to use 24 age- and sex-matched healthy students
LINE 216 and LINE 224: Authors should use numbers to indicate the timespan.
LINE 220: Abbreviation of calf serum is BCS. If authors use BSA, they should indicate "Bovine Serum Albumin (BSA)".
LINE 223: Percentage should be written with the symbol %.
LINE 224: The right name should be Coomassie Brilliant blue R 250. Furthermore, why authors stained the gel with Coomassie Brilliant Blue? They should justify this step.
LINE 230: How did the authors collect the data? Which instrument did they use?
LINE 251: It should be "WERE added to"
LINE 317: Authors reported that a total of 10,000 students were investigated, but only 46 students were considered to be students who repeat middle school. Is it possible that the students who repeat middle school are only 0.46% from 26 schools? Authors should better clarify and justify this point in the Materials and Methods section.
LINE 343: Figure 3? It should be Figure 1.
LINE 343 (Figure 3, probably Figure 1): As reported in Materials and Methods, the authors used the immunoblotting assay. How did they calculate pmol/g of 3-TH from the immunoblotting results? The authors have to declare this in 2.4 section. Furthermore, how is it possible that values are characterized by the same standard deviation?
LINE 354: Figure 1? It should be Figure 2
LINE 369: Figure 2? It should be Figure 3
LINE 369: Standard deviation is the same between Pre-CFS, Pre-EI, Post-CFS and Post-EI samples. Authors should justify this low data variability. A table to show Figure 2 data will be more appropriate compared to data graphical representation.
TABLE 2 and TABLE 3: RT and mz are probably referred to as Mean and Mean values, respectively. The authors should well specify this in the table caption. Furthermore, authors should also report the standard deviations or the standard errors of mean values.
LINE 426: How did the authors demonstrate the amelioration of mental fatigue? They should also report here (very briefly) in the discussion.
LINE 436: Authors should cite their results, reporting the related table or figure.
LINE 481: Authors should cite the reference at the end of the sentence.
LINE 524: Authors should cite the reference (Baklund et al., 2021) at the end of the sentence.
LINE 550: Authors have to cite a reference.
DISCUSSION: Authors should better discuss the limits of their study: for instance, the not high numerosity of the samples. Furthermore, discussing the results authors have to insert the reference to the figure or table present in the Results section.
Round 2
Reviewer 1 Report
The authors have done an absolutely wonderful job addressing almost all of my concerns. In fact, in many instances, their revisions exceeded my expectations. However, they have not addressed my major revision #10:
10. Figure 6 and related discussion: In the pathway analysis figure between HC and CFS, pathways B, C and D are significant, but they have very little pathway impact (~0.2). Similarly, in the pathway analysis figure between EI and CFS, pathways A, B, and C are marked even though they are not significant nor do they have an appreciable impact score (~0.12). However, the authors make much discussion of these findings and suggest that pathways are the underlying mechanisms of the intervention, but that is not supported by their data. The authors should mention in their results that most of the detected pathways were non-significant and the ones that were had very low to negligible impact. Relatedly, the authors should rewrite their discussion of those affected pathways as well.
Instead of addressing this, they have provided a long discussion of node centrality which, ultimately, is not a rebuttal to my point and in fact agrees with my initial criticism. Regardless of computational method, the pathway impact scores observed in the current study are, by all conventional standards, of negligible to low importance. Irrespective of significance and node radius, the impact on these pathways was low (again, between 0.1-0.2). Usually, pathways with p < 0.05 and impact scores > 0.5 are highlighted. Keep in mind, you are performing a predictive functional analysis of an unobserved state (namely, the pathway), so it is imperative you consider only those significant pathways with appreciable impact to avoid false inferences. So please, highlight this fact in the discussion and temper some of the more reaching claims made in the discussion.
